# Drug Safety in Translational Paediatric Research: Practical Points to Consider for Paediatric Safety Profiling and Protocol Development: A Scoping Review

**DOI:** 10.3390/pharmaceutics13050695

**Published:** 2021-05-11

**Authors:** Beate Aurich, Evelyne Jacqz-Aigrain

**Affiliations:** 1Department of Pharmacology, Saint-Louis Hospital, 75010 Paris, France; beate.aurich@gmx.fr; 2Paris University, 75010 Paris, France

**Keywords:** paediatric, clinical trial as topic, translational medical research, pharmacovigilance, clinical trial protocol as topic

## Abstract

Translational paediatric drug development includes the exchange between basic, clinical and population-based research to improve the health of children. This includes the assessment of treatment related risks and their management. The objectives of this scoping review were to search and summarise the literature for practical guidance on how to establish a paediatric safety specification and its integration into a paediatric protocol. PubMed, Embase, Web of Science, and websites of regulatory authorities and learned societies were searched (up to 31 December 2020). Retrieved citations were screened and full texts reviewed where applicable. A total of 3480 publications were retrieved. No article was identified providing practical guidance. An introduction to the practical aspects of paediatric safety profiling and protocol development is provided by combining health authority and learned society guidelines with the specifics of paediatric research. The paediatric safety specification informs paediatric protocol development by, for example, highlighting the need for a pharmacokinetic study prior to a paediatric trial. It also informs safety related protocol sections such as exclusion criteria, safety monitoring and risk management. In conclusion, safety related protocol sections require an understanding of the paediatric safety specification. Safety data from carefully planned paediatric research provide valuable information for children, parents and healthcare providers.

## 1. Introduction

Paediatric translational research facilitates an integrated, multidisciplinary and multidirectional approach aiming to improve children’s health [1,2,3]. Clinical trials are an important component of such research [3]. The planning of paediatric trials is informed by nonclinical and clinical data [4]. However, missing information on paediatric pharmacokinetics (PK), pharmacodynamics (PD), and safety should be proactively addressed in the study protocol [4,5,6,7]. This may include the use of innovative trial designs [8,9].

An integral part of paediatric trials is the assessment of treatment related risks and their management [4,5,10,11,12]. The collection and analysis of safety data as well as pharmacovigilance and risk management of paediatric trials are guided by the current understanding of the paediatric safety specification of the study drug or intervention [4,6,7,11,13]. Safety data evolves continuously. It may originate from a variety of data sources (e.g., in vitro studies, modelling and simulation, clinical trials, spontaneous reports, pharmacoepidemiology) [6,7]. The paediatric safety specification uses all available data to describe identified and potential treatment related risks and missing safety information [4,6,7].

Treatment response in children differs from adults and between different paediatric age groups [14,15,16,17]. Therefore, it is important to ensure that drug safety related protocol sections are based on current, age group specific safety specifications [4,5,7]. This informs, for example, case report form (CRF) design, informed consent/assent, risk management, pharmacovigilance, and planning of the resources needed for safety data analyses (e.g., programming) [4,5,18]. Whilst there is consensus that safety data for children is often lacking, the practical aspects of paediatric drug safety in the context of clinical research should be considered [19].

Thus, the objectives of this scoping review of the literature were to identify and summarise practical points to consider on how to establish a paediatric safety specification and how this information is subsequently built into a paediatric protocol.

## 2. Methods

PubMed, Embase, Web of Science and websites of regulatory authorities (EMA, FDA) and learned societies (CIOMS and WHO) were searched for publications, in English up to 31 December 2020, describing how a paediatric safety specification is developed and how it informs paediatric protocol development (Appendix A Table A1).

Titles and abstracts were screened and full-text articles were reviewed where applicable. In a second step, information from health authorities and learned societies relating to pharmacovigilance, risk management and protocol development was analysed.

## 3. Results

A total of 3534 publications were identified. After removing 54 duplicates, 3480 publications were screened (Appendix A Figure A1). No publication was identified providing practical advice on how to develop a paediatric safety specification and how this informs a paediatric protocol. Relevant information was found in a variety of, mostly regulatory, guidance documents. Therefore, information from health authorities, learned societies and the specifics of paediatric research were merged. The methodology for developing a paediatric safety specification and how it fits into the larger picture of paediatric drug safety research and risk management was summarised (Figure 1).

General and population specific guidance documents were combined to illustrate how the safety data of the study drug and the specifics of the paediatric study population can be used to develop a paediatric safety specification (Figure 2) [4,5,6,10,11,12,19,20,21,22,23,24,25,26,27,28].

Based on existing general and paediatric guidance documents, issued notably by CIOMS, the EMA and FDA, a figure was developed showing how the paediatric safety specification informs risk management and pharmacovigilance in a paediatric protocol (Figure 3) [4,5,6,10,12,23,24,26,28].

Since information on child specific challenges for the development of a paediatric safety specification are scattered in a considerable number of, mainly regulatory documents, a checklist was developed summarising points to consider for developing a paediatric safety specification (Table 1) [4,5,6,10,11,12,19,20,21,22,23,24,25,26,27,28].

Many regulatory guidance documents provided cross-references to related guidelines. Thus, a figure was developed to illustrate this interrelatedness for paediatric drug safety research. It shows how the paediatric safety specification, pharmacovigilance, risk management, paediatric trials, and clinical practice form part of a continuous cycle improving the understanding and management of the treatment related risks in children (Figure 4) [4,5,6,10,12,23,24,27,28].

Finally, the sections relating to drug safety in the EMA’s ICH E6 protocol guidance were reviewed [11]. Points to consider for the development of the safety sections of paediatric protocols were added. Examples of published paediatric studies were included to illustrate how the ICH E6 guidance can be translated into paediatric protocols (Table 2).

## 4. Discussion

The conduct of clinical trials is framed by regulatory guidelines which are mainly based on recommendations by international experts and concern mostly research in adults [65]. However, these guidelines are written for pharmaceutical companies and are usually not indexed in PubMed, Embase or Web of Science [11].

Whilst all clinical research needs to be conducted according to regulatory guidelines such as Good Clinical Practice (GCP), some of the regulations applying to drug development in pharmaceutical industry may not apply to independent academic research. For example, academic researchers do not need to submit a clinical study report to regulatory authorities. However, the underlying ethical and scientific reasonsing of many regulations are relevant for all researchers because they essentially go back to the notion of protecting study subjects and informing medical practice [7,11,63,66]. Thus, many apply to any clinical study in children regardless of sponsorship (i.e., pharmaceutical industry, academic institutions, or research consortia) [10,11,12,19,20,21,22,23,24,25,26,29,63,66].

The complexity of these regulations, lack of indexing, combined with limited practical guidance for protocol development from a paediatric drug safety perspective are challenging for paediatric researchers [19]. This article intends to provide a bridge between regulatory guidelines and the practical aspects of developing evidence-based paediatric protocol sections concerning drug safety. Therefore, guidelines from regulatory and learned society publications were combined with the specifics of the paediatric population [4,5,6,10,12].

Many drugs studied in children may have already been investigated in adults and sometimes in certain paediatric subpopulations [12]. Therefore, data will be available on nonclinical and clinical safety. It is however important to acknowledge that the interpretation of nonclinical studies for a paediatric population can be challenging [20,21]. Similarly, adult data or data from other paediatric subpopulations (e.g., other paediatric age groups), can only be used as a guide to focus safety data collection and analysis. It is recognised that adverse drug reactions (ADRs) observed in adults and other paediatric age groups may differ from those in the study population [14,15,16,17]. An ADR has been defined as a response to a drug which is noxious and unintended [67].

At the time of protocol development, the paediatric safety specification provides the necessary evidence for sections related to drug safety (Figure 3 and Table 2) such as the benefit-risk balance, exclusion criteria, rules for stopping treatment and safety monitoring [4,5,6,7,10,11,12]. It provides the rationale for the CRF design, management of treatment related risks, safety data collection and analysis [4,5,6,10]. Paediatric safety profiling takes into account the specifics of the paediatric study population, such as the epidemiology of the disease, pharmacokinetics (PK), pharmacodynamics (PD), risk factors for ADRs and adverse outcome, potential developmental toxicity and long-term risks, medication errors, comorbidities and comedications [4,5,6,12]. Information on pharmacogenomic risk factors and their distribution in different ethnicities can be helpful in assessing if and how these may change throughout childhood and whether genetic testing may need to be implemented at the time of trial inclusion [23,25,68]. Advanced techniques such as modelling and simulation, systems biology, machine learning and mathematical modelling can be used to study potential on/off target ADRs in children [12,69]. Frequent challenges for paediatric trials include the lack of age group specific PK/PD data, the need to have a formulation adapted to the study population and limited information on how short and long-term outcome can be assessed.

Using a structured approach to data collection and review provides clarity on identified and potential treatment risks and which safety information is missing (Figure 2 and Table 1). This will help deciding which safety data should be collected. Paediatric trials are a unique opportunity for high-quality safety data collection which should not be missed.

The paediatric safety specification also informs the risk management plan (RMP), which in turn provides a systematic, evidence-based approach to risk management and pharmacovigilance in paediatric clinical trials, observational studies and clinical practice [4,5,6,10]. RMPs are mandatory documents for pharmaceutical companies applying for a marketing authorisation in the European Union [5]. They include all safety relevant information and have a specific section for children [4,5]. The RMP also includes a plan on how missing safety information will be collected and the assessment of the effectiveness of risk management activities [4,5]. Similar to the safety specification, the RMP is a living document [5,6,10]. It is modified with each update of the safety specification. Thus, with the results of each new paediatric study, the paediatric safety specification may be updated, which in turn may be followed by an update of the RMP, the protocols of ongoing and planned paediatric studies as well as clinical practice (Figure 4).

In summary, the development of a paediatric protocol is informed by first developing a paediatric safety specification and then a risk management plan. These provide the necessary evidence for safety related protocol sections. Both are reviewed regularly and modified if needed. As new data become available, they may be updated and the sponsor will decide whether safety related protocol sections need to be amended [18]. Paediatric drug safety research is an integral part of clinical practice and paediatric research activities (Figure 4). It is a multidisciplinary effort that often begins prior to first-in-man studies and continues throughout the life cycle of a product [5,6,7,10]. The aim is to continue to provide children with medicines which have a favourable benefit-risk balance.

## 5. Conclusions

Paediatric drug safety in translational research is a collaborative effort of a multidisciplinary team throughout the life cycle of a medicinal product or device. Paediatric safety profiling requires an understanding of identified and potential risks as well as missing safety information in children. This informs relevant protocol sections reducing and managing treatment related risks, monitoring patients for these risks, and collecting data on frequency, severity, seriousness, and outcome of these risks. In addition, it informs plans for data collection in the CRF. Paediatric safety specification also helps to decide if any preclinical or PK studies are needed prior to conducting a trial, and which additional translational research may be warranted to improve children’s health and enhance the understanding of the benefit-risk balance of the study drug in children.

## Figures and Tables

**Figure 1 pharmaceutics-13-00695-f001:**
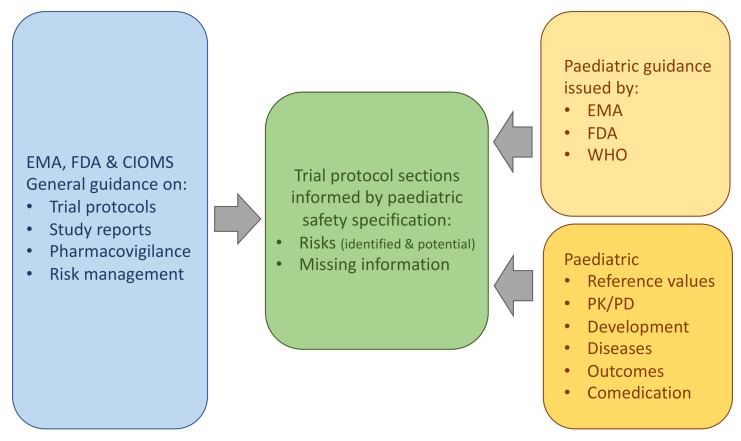
Paediatric research protocols and safety profiling. Process of combining information from health authorities, learned societies and the paediatric population. EMA—European Medicines Agency; FDA—Food and Drug Administration; CIOMS—Council for International Organizations of Medical Sciences; WHO—World Health Organization; PK—Pharmacokinetic; PD—Pharmacodynamic.

**Figure 2 pharmaceutics-13-00695-f002:**
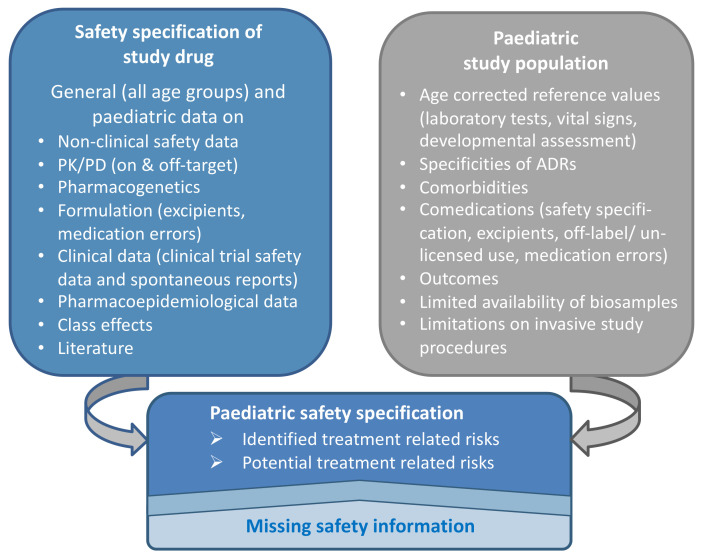
Data informing the paediatric safety specification. ADR—Adverse drug reaction; PK—Pharmacokinetic; PD—Pharmacodynamic.

**Figure 3 pharmaceutics-13-00695-f003:**
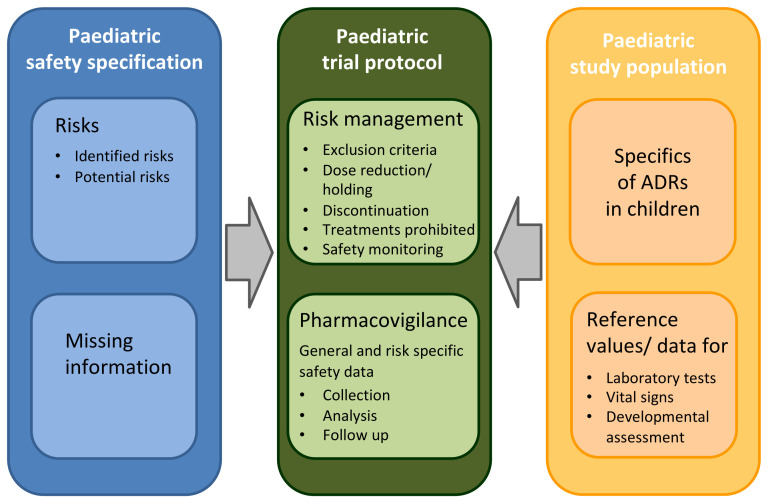
Relationship between paediatric safety specification and protocol development. ADR—Adverse drug reaction.

**Figure 4 pharmaceutics-13-00695-f004:**
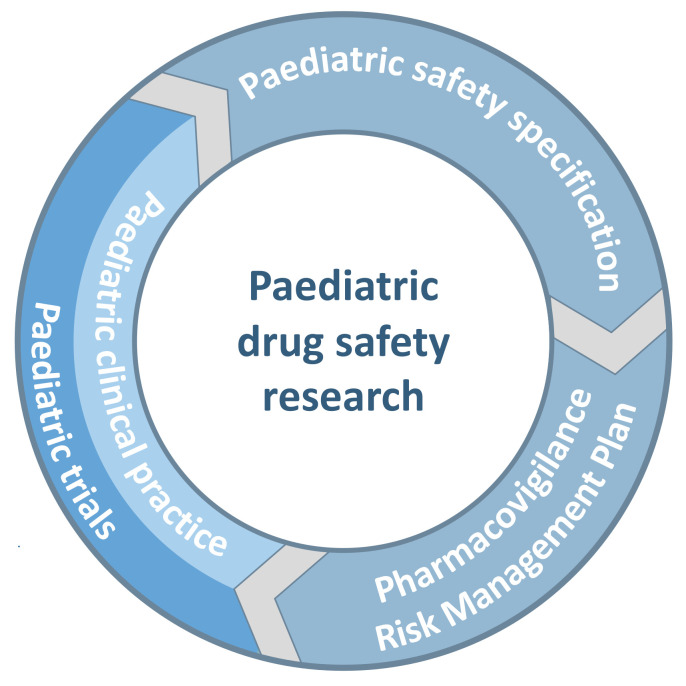
Paediatric drug safety research—Interaction between the paediatric safety specification, pharmacovigilance, risk management plan, clinical studies, and clinical practice.

**Table 1 pharmaceutics-13-00695-t001:** Proposal for a checklist for the development of a paediatric safety specification (not exhaustive).

Points to Consider for the Development of a Paediatric Safety Specification		Data Available for *			
		Adults	Children But Not Age Group of Study Population	Age Group of Paediatric Study Population	No Data
***Is there safety data on the study drug?*** ^†^					
**Nonclinical safety**	Consider assessment of developmental toxicity [20,21]				
**Pharmaco-** **kinetics (PK)**	If there is insufficient paediatric data, consider doing a PK study prior to an efficacy/safety study [22,23,29]				
**Pharmaco-** **dynamics (PD)**	Consider the feasibility of a paediatric PD study (on & off- target) prior to an efficacy/safety study and/or include PD into the study protocol [22,24,29]				
**Pharmaco-** **genetics**	Consider potential effect of age group specific pharmaco-genetics [25]				
**Formulation**	Ensure the formulation is adapted to the paediatric study population [24,26]				
**Excipients**	Consider formulation without excipients; assess potential excipient toxicity and total daily excipient dose (study drug(s) and comedications [24,26,30]				
**Medication** **errors**	Consider assessing risk for medication errors (study drug(s) and comedications) [5]				
**Clinical trial safety data**	Assess cumulative clinical trial safety data for all age groups and stratified by age group [4,5,6,10,12,27]				
**Spontaneous reports**	Summarise cumulative spontaneous reports for all age groups and stratified by age group [4,5,6,10,12,27]				
**Pharmaco-** **epidemiolo-gical data**	Explore whether paediatric drug utilisation and safety data are available for the study drug. Be aware of the specific challenges for paediatric pharmacoepidemiological data [31,32,33,34,35,36,37,38,39]				
**Class effects**	Review class effects in overall population and stratified by age group [4,5,6,10,12,27]				
**Literature search**	Search for safety concerns (i.e., signals) for the study drug/drug class for all age groups and children; review health authority websites [4,5,6,10]				
***Is there data on the paediatric study population?*** ^‡^					
**Age corrected reference** **values**	Be aware of the age group specific references values for laboratory data, vital signs and developmental assessment and how these change in the developing child [40,41,42,43,44,45,46]	**NA**			
**Specifics of ADRs in** **children**	Understand how adverse drug reactions (ADRs) present in children and how they differ from other paediatric age groups and adults [4]	**NA**			
**Paediatric comorbidities**	Describe common paediatric comorbidities for the study population (e.g., epilepsy and developmental delay) [12]	**NA**			
**Comedication**	Different compared to adults; increased risk of medication errors from comedication; risk of additive effect of excipients (total daily dose); consider the paediatric safety specification of comedications [12,24,26]	**NA**			
**Overall** **outcome**	Normal or disease specific development (e.g., Downs) [12,27,45]	**NA**			
**Limited** **biosampling**	Consider scavenged and opportunistic sampling; different recommendations on allowed blood volume by age & region [12,47]	**NA**			
**Limitations for study** **procedures**	Example: Blood pressure (BP) measurements in premature babies can be very disruptive. Small children will not sit still; consider taking BP during sleep [12,27]	**NA**			

* Check all that apply. ^†^ Useful for risk management (e.g., exclusion and stopping criteria, safety monitoring), pharmacovigilance (e.g., description of expected ADRs), safety data collection and analysis and benefit-risk assessment. ^‡^ Useful for case report form (CRF) alert values, design of study procedures for safety data collection and safety data analyses.

**Table 2 pharmaceutics-13-00695-t002:** EMA ICH E6 Protocol Guidance: Examples of the paediatric safety specification informing paediatric protocol development.

EMA ICH E 6 Protocol Guidance [11]		Points to Consider for the Development of Safety Sections in a Paediatric Protocol	Practical Examples in the Literature
Protocol Sections Concerning Drug Safety	ICH Guidance		
**Background information**	Summary of nonclinical studies that potentially have clinical significance and from clinical trials relevant to the trial	Including nonclinical safety data on excipients and missing, non-clinical information (e.g., juvenile animal studies) [4,5,20,21,30,48]	[48,49]
	Summary of the known and potential risks and benefits	Including risks of excipients and missing safety data (incl. medication errors) [4,5,22,24,26,30,48]	[50,51,52]
	Description and justification for route of administration, dosage, dosage regimen, and treatment period(s)	Age group specific PK/PD data for study drug and excipient; formulation adapted to paediatric target population [12,22,23,24,26,27,30,48,53]	[48,54,55]
	Description of the population to be studied	Describing how ADRs present in children (including risk factors and confounders), comedications (including excipients and medication errors) and paediatric reference ranges (laboratory tests, vital signs, development) [4,5,6,7,28]	[54,55,56,57,58,59]
	Literature and data relevant to the trial providing background for the trial	Including class effects in children [4,5,6,7]	[50,58,60,61]
**Objectives and purpose**	A detailed description of the objectives and the purpose of the trial	Safety should be a study objective (paediatric safety data is often lacking and reporting SAEs is mandatory) [4,5,6,7,19,28]	[49,54,55]
**Trial design**	A description of the stopping rules or discontinuation criteria for individual subjects, parts of trial and entire trial	Based on paediatric safety specification [4,7,10,11] Rules based on paediatric reference ranges (laboratory tests, vital signs, development) [4,7,10,28]	[56,62]
**Selection and withdrawal of subjects**	Subject exclusion criteria	Based on paediatric safety specification [4,7,10,11] Contraindications for children [4,7,10,11] Rules based on paediatric reference ranges (laboratory tests, vital signs, development) [4,7,10,28]	[51,55,57,58,60]
	Withdrawal criteria and procedures specifying when and how to withdraw subjects, type and timing of data collection and FU	Based on the paediatric safety specification [4,7,10,11] Rules based on paediatric reference ranges (laboratory tests, vital signs, development) [4,7,10,28]	[55,56,60]
**Treatment of subjects**	FU period(s) for subjects	Safety FU: based on paediatric safety specification [4,7,10,11]	[60]
	Treatment(s) not permitted for safety reasons before/during the trial	Based on paediatric safety specification [4,7,10,11]	[54]
**Assessment of safety**	Specification of safety parameters	Based on paediatric safety specification [4,7,10,11] Parameters based on paediatric reference ranges (laboratory tests, vital signs, development) [4,7,10,28]	[55,59,60]
	The methods and timing for assessing, recording, and analysing safety parameters	Based on the paediatric safety specification, including collection of missing paediatric safety information [4,7,10,11]; may be restricted by limitations for biosampling and invasive testing; physical examinations need to be adapted to children [4,12,19,23,27,63] Record medication errors for the study drug and comedications in the CRF [4,12,19,23,27] Record comedications in the CRF using paediatric standards (i.e., mg/kg) [4,7,10,12] Adapt safety data analyses to the paediatric age group; certain data analyses should only be done by paediatric specialists (e.g., ECGs) [4,7,10,11]	[50,56,58,59,61]
	Procedures for eliciting reports and for recording and reporting AEs	How do AEs/ADRs present in children, comorbidities [4,7,10,12]	[54]
	Type and duration of FU after AEs	Based on paediatric safety specification, including missing information [4,5,7,10,11]	[60]
**Statistics**	Statistical methods (including timing of interim analyses)	Based on paediatric safety specification, including missing safety information [4,5,7,10,11]	[55,56,64]
	Safety criteria for trial termination	Based on paediatric safety specification [4,7,10,11]	[55,56]

ADR = Adverse drug reaction; AE = Adverse event; CRF = Case Report Form; ECG = Electrocardiogram; EMA = European Medicines Agency; FU = Follow-up; ICH = International Council for Harmonisation of Technical Requirements for Pharmaceuticals for Human Use; PD = Pharmacodynamic; PK = Pharmacokinetic; SAE = Serious Adverse Event.

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
