# Peer review of "Drug Safety in Translational Paediatric Research: Practical Points to Consider for Paediatric Safety Profiling and Protocol Development: A Scoping Review"

_pharmaceutics, 2021, doi:10.3390/pharmaceutics13050695_

Round 1

Reviewer 1 Report

This manuscript reviews the current literature to provide evidence-informed approaches to a study strategy for paediatric drug studies, an important consideration given that typically safety pharmacology plans are drawn up for adult studies.  The search and review strategy are described and are appropriate.  The conclusions are drawn from the data collected. 

I am not quite sure of the point of the multiple fleur de lis on Table I as it seems that all of the boxes are checked. 

In terms of pharmacogenetics/genomics are there relevant considerations beside age?  As an example, the ultrarapid metabolizer phenotype of CYP2D6 is significantly higher among children of Middle Eastern and South Indian descent than children of Northern European ancestry, and this has been correlated with a difference in risk for CYP2D6 activation mediated events.

As a minor point ADRs are mentioned but not defined.  It might be helpful given the premise of the paper if the authors define what in their view constitutes an adverse drug reaction.

Author Response

Dear Reviewer 1,

Thank you for your very helpful review and comments. Please find below our responses to your comments:

Comment 1:

“This manuscript reviews the current literature to provide evidence-informed approaches to a study strategy for paediatric drug studies, an important consideration given that typically safety pharmacology plans are drawn up for adult studies. The search and review strategy are described and are appropriate. The conclusions are drawn from data collected.

I am not quite sure of the point of the multiple fleur de lis on Table I as it seems that all of the boxes are checked.”

Response: This maybe a placeholder added during the editing process. We have removed them.

Comment 2:

“In terms of pharmacogenetics/genomics are there relevant considerations beside age?  As an example, the ultrarapid metabolizer phenotype of CYP2D6 is significantly higher among children of Middle Eastern and South Indian descent than children of Northern European ancestry, and this has been correlated with a difference in risk for CYP2D6 activation mediated events.”

Response: The interactions between pharmacogenetics/ genomics and child development remain underexplored. We agree that factors independent of child development/ age are important when considering pharmacogenetic risk factors for adverse drug reactions. We have added a short comment on this in the discussion (lines 410 to 413) with several references in case readers are interested to further explore this topic:

Information on pharmacogenomic risk factors and their distribution in different ethnicities can be helpful in assessing if and how these may change throughout childhood and whether genetic testing may need to be implemented at the time of trial inclusion [23,25,68].

Comment 3:

“As a minor point ADRs are mentioned but not defined.  It might be helpful given the premise of the paper if the authors define what in their view constitutes an adverse drug reaction.”

Response: We have added the EMA’s definition and the corresponding reference in the discussion (lines 399 to 401):

An ADR has been defined as a “response to a drug which is noxious and unintended”[67].

Reviewer 2 Report

The manuscript entitled " Drug safety in translational paediatric research: Practical points to consider for paediatric safety profiling and protocol development – A scoping review" by Aurich B. et al presents a review about a proposal to establish pediatric safety specifications for study protocols.

The manuscript is well written and adds and provides informations from several international guidelines, consensus, and scientific articles related to the subject.

I have a few comments that may improve the current version of the manuscript:

1- Please check the grammar and spelling of teh following sentences and words:

a- " Safety data evolves continuously and may originate from a variability of.." Please re-word

b- " Pubmed, Embase, Web of Science and websites of regulatory authorities (EMA, 58 FDA) and learned societies (CIOMS and WHO) were searched for publications in English 59 up to 31/12/2020 .." Adequate the format of the date.

c- " Since information on child specific challenges for the development of a paediatric safety specification are scattered in a considerable number of, mainly regulatory, documents, ..." Is there a comma or a typo?

d- Points to consider for the development the safety sections of .." Please, check the sentence.

e- Table 1: Pharmacokinetics  , pharmacodynamics, pharmacogenetics

2- From Table 1. Consider a paediatric PD study (on & off-target) prior to an efficacy/ safety study and/or include PD into the study protocol. Do the authors think this is feasible and/or practical for all indications without pediatric safety data?

3- Perhaps it would be interesting to explore new approaches such as using system biology, expression profiles, and other mathematical approaches that may aid to model the on/off target effects. Could the authors comment on this topic?

4- I do not think it would be adequate to suggest performing new studies if there is enough information about the formulation excipients even in a pediatric population with the same age of the study population.

5-It would be helpful for the reader to have the description of an example about how the authors propose to include, and which specific information based on Tables 1 and 2, for the development of the safety sections of a pediatric protocol.

Author Response

Dear Reviewer 2,

Thank you for your very helpful review and comments. Please find below our responses to your comments:

Comment 1: 

The manuscript entitled “Drug safety in translational paediatric research: Practical points to consider for paediatric safety profiling and protocol development – A scoping review” by Aurich B. et al presents a review about a proposal to establish pediatric safety specifications for study protocols.

The manuscript is well written and adds and provides information from several international guidelines, consensus, and scientific articles related to the subject.

I have a few comments that may improve the current version of the manuscript:

Please check the grammar and spelling of the following sentences and words:

a- " Safety data evolves continuously and may originate from a variability of.." Please re-word

Response: We have rephrased this section to the following, braking it into several shorter sentences (lines 41 to 43):

Safety data evolves continuously. It may originate from a variety of data sources (e.g. in vitro studies, modelling and simulation, clinical trials, spontaneous reports, pharmacoepidemiology).

b- " Pubmed, Embase, Web of Science and websites of regulatory authorities (EMA, 58 FDA) and learned societies (CIOMS and WHO) were searched for publications in English 59 up to 31/12/2020 .." Adequate the format of the date.

Response: We have modified the formatting of the date to 31 December 2020 (line 67):

Pubmed, Embase, Web of Science and websites of regulatory authorities (EMA, FDA) and learned societies (CIOMS and WHO) were searched for publications in English up to 31 December 2020 describing how a paediatric safety specification is developed and how it informs paediatric protocol development.

c- " Since information on child specific challenges for the development of a paediatric safety specification are scattered in a considerable number of, mainly regulatory, documents, ..." Is there a comma or a typo?

Response: Yes, this is a typing error. There should be no comma between regulatory and documents (line 103):

Since information on child specific challenges for the development of a paediatric safety specification are scattered in a considerable number of, mainly regulatory documents, a checklist was developed summarising points to consider for developing a paediatric safety specification (Table 1) [4-6,10-12,19-28].

d- Points to consider for the development the safety sections of .." Please, check the sentence.

Response: We have added the missing word to this sentence (line 360):

Points to consider for the development of the safety sections of paediatric protocols were added.

e- Table 1: Pharmacokinetics, pharmacodynamics, pharmacogenetics

Response: We noted that the automatic hyphenation tool had been activated. We have inactivated it and corrected manually where necessary.

Comment 2:

From Table 1. Consider a paediatric PD study (on & off-target) prior to an efficacy/ safety study and/or include PD into the study protocol. Do the authors think this is feasible and/or practical for all indications without pediatric safety data?

Response: We do not think it is feasible and/or practical to conduct paediatric PD studies for all indications without pediatric safety data. Whilst indication specific safety PD data is useful, we feel that existing safety PD data for the same pediatric age group in a different indication might be sufficient. In addition, advanced techniques such as modeling and simulation and bridging for drug safety from other pediatric populations can be used. We have added the following to the line on PD in Table 1:

Consider feasibility of a paediatric PD study (on & off-target) prior to an efficacy/ safety study and/or include PD into the study protocol

Comment 3:

Perhaps it would be interesting to explore new approaches such as using system biology, expression profiles, and other mathematical approaches that may aid to model the on/off target effects. Could the authors comment on this topic?

Response: We have added a comment in the discussion and two references (lines 413 – 416):

Advanced techniques such as modelling and simulation, systems biology, machine learning and mathematical modelling can be used to study potential on/off target ADRs in children [12,69].

Comment 4:

I do not think it would be adequate to suggest performing new studies if there is enough information about the formulation excipients even in a pediatric population with the same age of the study population.

Response: We agree with you. We could not find any section in our article suggesting this. Could you please clarify to which line or sentence you are referring to? Thank you.

Comment 5:

It would be helpful for the reader to have the description of an example about how the authors propose to include, and which specific information based on Tables 1 and 2, for the development of the safety sections of a pediatric protocol.

Response: This information is included in Table 2. We have modified the wording of the table headers to clarify:

First column -          Protocol sections concerning drug safety

Second column -     ICH Guidance

Third column-         Points to consider for the development of safety sections in

                                 a paediatric protocol

Fourth column -      Practical examples in the literature